# The European Union Is Still Unable to Find Nemo and Dory-Time for a Reliable Traceability System for the Marine Aquarium Trade

**DOI:** 10.3390/ani11061668

**Published:** 2021-06-03

**Authors:** Monica V. Biondo, Ricardo Calado

**Affiliations:** 1Fondation Franz Weber, 3011 Bern, Switzerland; 2ECOMARE, CESAM—Centre for Environmental and Marine Studies, Department of Biology, Santiago University Campus, University of Aveiro, 3810-193 Aveiro, Portugal

**Keywords:** FAIR data, marine ornamental fishes, marine aquarium trade, TRACES

## Abstract

**Simple Summary:**

The EU is one of the main markets for marine ornamental species. Their entrance into the EU, as well as their circulation between member states, is supposed to be highly regulated. Surprisingly, it is currently impossible to answer simple questions, such as how many Nemos and Dorys are imported each year into the EU, or where do they come from? This lack of knowledge is difficult to understand, as all these organisms enter the EU by air-shipping and must be controlled at customs offices in international airports. This scenario favors “business as usual” and does not allow to verify the claims on sustainability commonly made by the marine aquarium industry. However, the EU already operates a platform that may allow to collect such information in a reliable way and shed light on this blurry industry, the Trade Control and Expert System (TRACES). This platform can start by surveying marine ornamental fishes, so the EU can finally know how many Nemos and Dorys are being imported and where they are sourced from. If this approach works, marine ornamental invertebrates can also be monitored, and reliable databases can finally be assembled to document the marine aquarium trade in the EU.

**Abstract:**

The EU is one of the main importers of marine ornamental species sourced from tropical coral reefs around the world. While the entrance of live organisms into the EU, along with their intra-EU circulation, is framed within stringent control mechanisms, to date, no reliable figures exist concerning which marine ornamental species are imported, in what numbers, and where they are sourced from. This lack of reliable data in the EU on the trade of marine ornamental species is puzzling if one considers that all these imported specimens must be controlled at customs offices located in international airports. Such data deficiency favors the prevalence of blurry supply chains and a “business as usual” mindset that hampers any serious effort to promote sustainability in the marine aquarium industry. To safeguard the collection of findable, accessible, interoperable, and reusable (FAIR) data, we suggest that the EU platform Trade Control and Expert System (TRACES) refines its surveillance on the trade of marine ornamental species. The detailed survey of marine ornamental fishes alone can be used as a proof of concept to validate the use of TRACES for this purpose and, if successful, it can later be expanded to all other taxonomic groups of marine ornamental species.

## 1. Introduction

The marine aquarium trade is a global-scale industry that supplies enthusiastic hobbyists, along with public and private aquariums, with a multitude of vertebrate and invertebrate organisms that mostly originate from coral reefs in southeast Asia [1]. Commonly being referred as a multi-million dollar activity, the fact is that figures on the value of this industry are commonly outdated [1,2]. Indeed, accurate and reliable sources of information on this important activity are either missing or unavailable. Unfortunately, it is not only the value pertaining to this industry that is lacking reliable and up to date figures. The volume, taxonomy, and geographic origin of the live specimens being traded are also largely unknown [2,3]. While some works have tried to estimate these figures for importing markets, such as the USA [4,5], Australia [6], and Europe [2,7,8,9], they all acknowledge limitations (e.g., taxonomy and numbers of specimens being traded are largely unknown) and consider that the figures presented are most likely underestimates.

With the EU often being reported as one of the main importing markets of marine ornamental species and championing stringent control mechanisms that monitor the entrance and circulation between member states and associated states of live organisms [8,10], it is puzzling that no reliable figures exist for such a well-established economic activity [2,7,8]. This fact is even more surprising if one acknowledges that most marine ornamental species, namely specimens being collected from the wild, enter the EU via air shipping, thus having as their entrance points highly monitored and regulated facilities, namely customs offices at international airports. At present, the EU largely ignores the overall value associated with the trade of marine ornamental species, the list of species being imported, their numbers, and their country of origin. The EU is therefore unable to find Nemo and Dory because no reliable traceability protocol is in place to monitor marine ornamental species once they enter the European market.

## 2. Blurry Supply Chains Allowing “Business as Usual” to Go On

The EU acknowledges the advantages of tracing all movements of live organisms in arriving to, moving between, and exiting its member states. This survey reduces the potential impacts of disease outbreaks and may allow authorities to trigger faster mitigation actions along trade chains. To do so, trade chains must be well-defined and one must be able to simultaneously trace the previous and subsequent link of any player in that chain. That is certainly not the case of the marine aquarium trade [3]. Trade chains associated with the importation of marine ornamental species have long been recognized as being diffuse and blurry, making it virtually impossible to pinpoint the number and accurate geographic origin of most specimens being traded [2,11,12]. Several specimens are collected in remote regions and pass over multiple middlemen until they arrive to an international wholesaler in the main cities of exporting countries served by international air flights (e.g., Philippines and Indonesia). These are the trade chain players that prepare marine ornamentals for international air-shipping to importing countries. For many specimens, the first time they will likely be recorded in a database is at the wholesale facility. However, such databases are not framed under FAIR data principles [13], as these data on marine ornamental species will not be readily findable (F), will likely only be accessible (A) to some corporate staff and authorities, will not be interoperable (I) (as these databases are often isolated data islands), and will not be reusable or re-used (R).

Not knowing the accurate place of origin of marine ornamentals being collected impairs any reliable assessment on the effect of the fishing effort targeting donor populations in the wild. How can claims on sustainability be performed by the marine aquarium industry, or any managing authority, under this scenario? Even if only employing sustainable collection practices, operations per se may not be sustainable.

The blurry nature of marine ornamentals supply chains is not only an issue when these species are being traded at domestic level before exiting their source country. At times, the country of origin being declared when marine ornamentals are entering the EU may not be the country where those specimens were collected. While some large sized wholesalers operating in the EU import marine ornamentals directly from source countries, other smaller players import them from third countries that act as logistics hubs (e.g., Singapore) that facilitate imports.

## 3. TRACES, the Right Tool to Monitor Marine Ornamental Species Trade in the EU

The most puzzling issue concerning why the EU continues to largely ignore what marine ornamental species are being imported, in what numbers, and where they originate from is that all member states already employ one of the most advanced tracking systems in the world that targets the import, export, and intra-EU trade of live animals, animal products, and plants—the Trade Control and Expert System (TRACES). TRACES is an EU multilingual online platform that was put forward to safeguard sanitary certification related with the import, export, or trade at intra-EU level of animals and plants [9]. EU law specifically refers that live animals, animal products, feed, and plants must be accompanied by official certificates that attest to their compliance with existing EU regulations. When live animals and plants are imported into the EU, as well as when they are traded within the EU single market, TRACES records all official controls, as well as the pathway of those organisms from their origin to their destination. This online platform is already used by 85 countries with over 40,000 users in the world, including several non-EU countries (e.g., Iceland, Norway, Switzerland…), is available in 34 different languages, being accessible 24 h a day, seven days a week, and is free of charge after registration [9]. Authorities in EU points of entrance, or at destination, are pre-notified and can plan controls in due time, including on animal welfare related issues. These features alone already make TRACES a very appealing platform to monitor the import of marine ornamentals. Additionally, most countries exporting these species to the EU also export other goods that are also monitored under TRACES (e.g., animal products and food) and, as such, are already familiar with the platform. According to the European Commission, TRACES aims “to streamline the certification process and all linked entry procedures and to offer a fully digitized and paperless workflow”. With a built-in statistical tool that allow users to extract data referring to imports into the EU, exports from the EU, and intra-EU trade, TRACES is the right tool to allow one of the largest markets of marine ornamental species in the world to gain an unprecedented insight on this illusive economic activity. This approach has already been previously suggested by Biondo [8,14], but is yet to be put into practice. Although some technical adjustments may have to be put into place to allow the collection of data relevant for monitoring the trade of marine ornamental species, these will allow the EU to perceive for the first time the sustainability of this economic activity using a science-based approach.

## 4. The Way Forward

TRACES was not specifically developed to monitor the trade of wildlife in the EU, although all wildlife legally entering the EU will somehow be recorded in this platform. While TRACES has distinguished freshwater from marine ornamentals since 2014, the diversity of traded species and number of imported specimens remains not fully reported. Moreover, the multitude of marine ornamental fishes and invertebrates traded for marine aquariums may be a phenomenal obstacle to tackle and likely impair any feasible solution to be implemented in the short-term to better understand what is imported, from where, and in what numbers. As such, to test the suitability of the solution here advocated, it is suggested that only marine ornamental fishes are covered in detail. This initiative will act as a proof of concept that, once validated, may latter be extended to some groups of marine ornamental invertebrates and ultimately cover the whole biodiversity of organisms supplying the marine aquarium industry.

Making the detailed reporting of marine ornamental fish species name, number, and origin mandatory by exporters to veterinary authorities at customs in international airports would allow TRACES to be the first platform able to generate FAIR data for one of the biggest importing markets of marine ornamentals in the world. This goal will only be achieved if the origin countries from which marine ornamental species are sourced from are only allowed to export these organisms to the EU if the species and number of specimens being traded, where have they been collected, and how are clearly indicated. Marine ornamental fishes originating from aquaculture should also be clearly labelled as such, to avoid a biased perception of the fishing pressure targeting the populations of those species in the wild. However, it must be safeguarded that data are reported in a way that serves the purposes of sustainability and conservation. Reporting species lists at higher taxonomic levels (including family or genus) would hamper the use of data to flag species being more prone to overfishing. The same caveat would also arise if the volume of traded species is reported in any other unit than number of specimens. Reporting as place of origin any other country than the one where fishes were originally captured from the wild would also impair the reliable use of such data.

With the marine aquarium trade in the EU being so speciose when compared to other economic activities that are also surveilled by TRACES, it is certainly important that the nomenclature employed is up to date or at least allows synonyms and alternative names so data recording can continue to be performed in a reliable way [8]. The reporting of species names should be made from dropdown menus where fish species being listed cannot be restricted solely to species already being traded. Such a procedure would overlook new species being recruited into the trade, and therefore the list provided should be comprehensive and cover all known fish species from tropical coral reefs. As already referred to above, it is paramount that the use of synonyms is allowed to accommodate taxonomic updates that are regularly performed by the scientific community.

## 5. Conclusions

Without a reliable way to compile sound data on species composition, numbers, and origin of marine ornamental species imported to the EU, any claims by the marine aquarium industry on its sustainability are based on assumption and not science-based facts. The EU may lead the first approach ever delivering FAIR data on marine ornamental species. EU member states have pledged their commitment to ocean conservation and coral reefs regeneration (see Mission Starfish 2030 [15]). Moreover, at the last Conference of the Parties to CITES (the Convention on International Trade in Endangered Species of Fauna and Flora) in 2019, designed to ensure sustainable international trade in threatened animals and plants species, the EU co-proposed to start scrutinizing the trade of marine ornamental fishes, with the proposal being supported by the 183 member states [16,17]. If TRACES is used to its full potential, not only will the EU be able to rapidly find out how many Nemos and Dorys are imported every year by member states, but also from where they originate. This active survey on the imports of marine ornamental species, starting with fishes and expanding to cover marine invertebrates, is most of all an effort to safeguard that Nemo and Dory may continue to occur in their natural habitats—coral reefs.

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
