# Peer review of "The European Union Is Still Unable to Find Nemo and Dory-Time for a Reliable Traceability System for the Marine Aquarium Trade"

_animals, 2021, doi:10.3390/ani11061668_

Round 1
Reviewer 1 Report
The manuscript by Biondo and Calado is a commentary on the use of the TRAde Control and Expert System (TRACES) to survey and monitor marine ornamental fish trade to and within EU members.
The marine aquarium trade is a highly profitable global industry, but has a series of flaws that hamper the traceability of the specimens collected. I imagine that, at least in part, this is a consequence of the use of non-sustainable methods and the collection above the maximum number of individuals allowed at a given location/country.
The use of TRACES will certainly improve the traceability and control of imported fish once they reach the EU, but I am not sure whether it will satisfactorily meet all the FAIR goals. For that to happen, detailed information on the number of fish collected, where and in which conditions they were collected are required and will only be available if the countries/companies responsible provide these information. Thus, the engagement of the countries of origin is essential for this framework to work on its entirety providing enforcement and control.
Reviewer 2 Report
The Commentary submitted by Monica Biondo and Ricardo Calado (animals-1233787) presents a clearly articulated and well thought-out perspective on the use of the TRAde Control and Expert System (TRACES) to monitor the quantity, diversity, and origin of marine ornamental fishes imported into the European Union. Both authors are respected experts on the topic, and their commentary nicely summaries the implications of their recent research. The commentary is also quite timely given that CITES is currently "scrutinizing" the trade of marine ornamental fishes.
I have no hesitation in recommending this article for publication.
Below are some minor comments the authors may wish to consider while finalising their article for publication.
General
There are some interesting parallels between the EU and Australia in their approach to monitor wildlife trades with the intent of managing biosecurity risks. The authors may glean some insights from the following article; certainly, the notion that the taxonomy of the specimens being traded is largely unknown also applies to Australia.
Trujillo-Gonzalez, A., Militz, T.A. 2019. Taxonomically constrained reporting framework limits biodiversity data for aquarium fish imported to Australia. Wildlife Research 46, 355-363.
Simple Summary & Abstract
The abstract is elegantly written and provides a clear summary of the authors' commentary. However, the Simple Summary currently reads as an inferior draft of the abstract. After having a look at the Instructions for Authors, I would recommend the authors treat the Simple Summary as the "Highlights" often required for other journals: three to four sentences briefly summarising the article's 'take-home' message.
Abstract: "FAIR data" - please define this acronym at first mention in abstract (findable, accessible, interoperable, reusable).
Section 2: "At times, the country of origin being declared when marine ornamentals are entering the EU may not always be the country where those specimens were collected" - "at times" and "not always" are redundant - suggest "always" is removed.
Section 4: "Making the detailed report of marine ornamental fish species name, number and origin mandatory would allow TRACES" - do the authors mean "reporting"?
Additionally, it is presently unclear for who the authors propose making detailed reporting mandatory. Will reporting be the responsibility of EU customs officials, the government of the exporting country, the commercial operator importing the marine organisms, or the commercial operator exporting the marine organisms to the EU? This needs to be clarified and, depending on the responsible party, consideration should be given to the cost responsibility for reporting.
Further ambiguity of responsibility stems from "all wildlife legally entering the EU will somehow be recorded in this platform" - is it not known how this information is recorded?
Section 5: "...the EU, co-proposed to start scrutinizing". - would suggest the comma is inappropriate here.
End
Reviewer 3 Report
To begin with, I would like to commend the Authors of the commentary entitled "The European Union is still unable to find Nemo and Dory - Time for a reliable traceability system for the marine aquarium trade" for bringing up the rather difficult and (scientifically) unpopular subject of ornamental fish trade. It is indeed an industry which has now for years evaded mainstream interest and therefore proper regulatory and legislatory measures, but it is rather a global issue, not only of the EU. Definitely, overfishing of ornamental species is a serious threat to the aquatic biodiversity, which is why the use of monitoring systems must be established, in order to provide at least some degree of institutional control.
The reviewed commentary was written using good English, I only have an objection regarding the very first sentence in the Introduction, there is something stylistically wrong with it. For instance, "namely" should be replaced with "mostly", but there is more to correct here.
I do not have any major objections regarding the manuscript, but I wish to provide some constructive criticism. I would like to give a few suggestions which the Authors may later use to build upon to expand their commentary, at least a little.
Firstly, considering the Authors' self-citations of previous papers, I understand that they focused their attention fully on marine ornamentals, as it appears to be their speciality. However, in my opinion, the subject of ornamental fish trade can not be be discussed without (briefly) mentioning freshwater species. There are notable differences, obviously, as the vast majority of traded freshwater ornamentals are artificially bred specimens, what remains in stark contrast to marine fish. Nevertheless, the ornamental fish trade is definitely dominated by freshwater fish, and even if the import of wild-caught batches of freshwater ornamentals is not as essential as the trade of captively-bred fish (within EU countries or imported/exported from the EU), the monitoring of this part of the ornamental trade should be perceived as an equally important task as it is for marine fish. Thus, I suggest that the Authors could at least briefly allude to the subject of freshwater fish trade and the importance of its control, likewise as for marine species. I give two exempliary citations below which may be helpful in this regard.
Evers, H. G., Pinnegar, J. K., & Taylor, M. I. (2019). Where are they all from?–sources and sustainability in the ornamental freshwater fish trade. Journal of fish biology, 94(6), 909-916.
King, T. A. (2019). Wild caught ornamental fish: a perspective from the UK ornamental aquatic industry on the sustainability of aquatic organisms and livelihoods. Journal of fish biology, 94(6), 925-936.
Secondly, the currently uncontrolled trade of (mostly) freshwater ornamental fish and shellfish is a gateway for the spreading of (potentially) invasive species, what may have serious consequences for local aquatic environments, such as the transmission of the crayfish plague. Once again, I give two citations as point of reference.
Chan, F. T., Beatty, S. J., Gilles Jr, A. S., Hill, J. E., Kozic, S., Luo, D., ... & Copp, G. H. (2019). Leaving the fish bowl: the ornamental trade as a global vector for freshwater fish invasions. Aquatic ecosystem health & management, 22(4), 417-439.
Svoboda, J., Mrugała, A., Kozubíková‐Balcarová, E., & Petrusek, A. (2017). Hosts and transmission of the crayfish plague pathogen Aphanomyces astaci: a review. Journal of fish diseases, 40(1), 127-140.
Thirdly, although the trade of GMO ornamental fish is not allowed within the EU, it is fairly easy to purchase such genetically-modified, fluorescent fish (such as the "GloFish" from the US) in various European countries. As the transgenic technology keeps on skyrocketing in recent years, it is reasonable to predict that the number of such genetically engineered ornamental species available on the market will only keep on growing, which is yet another argument for a better use of ornamental fish trade control tools.
Debode, F., Marien, A., Ledoux, Q., Janssen, E., Ancion, C., & Berben, G. (2020). Detection of ornamental transgenic fish by real-time PCR and fluorescence microscopy. Transgenic research, 29(3), 283-294.
Finally, the artificial breeding of marine ornamental species has seen some significant improvements in recent years (citation below), therefore it is highly possible that the inhouse production of these fish within the EU will grow in near future. It is also likely that GMO marine fish will soon be entering the market, as well. This shows that the need to control the production and trade within the EU will probably be at least as important as the monitoring of imports which arrive from all over the world.
Chen, J. Y., Zeng, C., Jerry, D. R., & Cobcroft, J. M. (2020). Recent advances of marine ornamental fish larviculture: broodstock reproduction, live prey and feeding regimes, and comparison between demersal and pelagic spawners. Reviews in Aquaculture, 12(3), 1518-1541.
I am sorry that this review is so lengthy, but I believe that my remarks will broaden the aspects mentioned in this paper. Simply put, there needs to be even more emphasis put on the importance of the monitoring of the ornamental fish trade in the EU, for a variety of reasons.
